# Global Longitudinal Strain in Cardio-Oncology: A Review

**DOI:** 10.3390/cancers15030986

**Published:** 2023-02-03

**Authors:** Grzegorz Sławiński, Maja Hawryszko, Aleksandra Liżewska-Springer, Izabela Nabiałek-Trojanowska, Ewa Lewicka

**Affiliations:** 1Department of Cardiology and Heart Electrotherapy, Medical University of Gdańsk, 80-210 Gdańsk, Poland; 2Cardio-Oncology Outpatient Clinic, University Clinical Center, 80-210 Gdańsk, Poland; 3Club 30, Polish Cardiac Society, 00-193 Warsaw, Poland; 4Department of Cardiology, Medical University of Gdansk, 80-210 Gdańsk, Poland

**Keywords:** cancer treatment, cardiotoxicity, echocardiography, speckle tracking echocardiography, strain

## Abstract

**Simple Summary:**

The assessment of global longitudinal strain (GLS) has an established role in cardio-oncology in the early diagnosis of the cardiotoxicity of anticancer treatments. Baseline left ventricular (LV) GLS and right ventricular (RV) GLS assessments can identify patients at risk for systolic dysfunction and heart failure due to the cardiotoxicity of various cancer treatments. Depending on the baseline risk of cardiotoxicity assessed before treatment initiation, serial echocardiography with a GLS assessment should be performed during the anticancer therapy to enable prompt initiation and dose adjustment for cardioprotection in the event of subclinical myocardial contractile dysfunction. We recommend routine GLS assessments in cardio-oncology patients if the patient’s imaging conditions allow it.

**Abstract:**

Several therapies used in cancer treatment are potentially cardiotoxic and may cause left ventricular (LV) dysfunction and heart failure. For decades, echocardiography has been the main modality for cardiac assessment in cancer patients, and the parameter examined in the context of cardiotoxicity was the left ventricular ejection fraction (LVEF). The assessment of the global longitudinal strain (GLS) using speckle tracking echocardiography (STE) is an emerging method for detecting and quantifying subtle disturbances in the global long-axis LV systolic function. In the latest ESC guidelines on cardio-oncology, GLS is an important element in diagnosing the cardiotoxicity of oncological therapy. A relative decrease in GLS of >15% during cancer treatment is the recommended cut-off point for suspecting subclinical cardiac dysfunction. An early diagnosis of asymptomatic cardiotoxicity allows the initiation of a cardioprotective treatment and reduces the risk of interruptions or changes in the oncological treatment in the event of LVEF deterioration, which may affect survival.

## 1. Introduction

Several therapies used in cancer treatment are potentially cardiotoxic and may cause left ventricular (LV) systolic dysfunction and heart failure (HF), principally anthracyclines but also HER2-targeted therapies, tyrosine kinase inhibitors, multiple myeloma therapies, MEK and RAF inhibitors, immune checkpoint inhibitors, and radiotherapy [1]. For decades, echocardiography has been the main mode of examination for the assessment of cardiac function in cancer patients. In the context of cardiotoxicity, the left ventricular ejection fraction (LVEF) has been assessed. However, the measurement of LVEF depends on hemodynamic conditions. Two-dimensional (2D) echocardiography depends on expertise, image quality, and assumptions of LV geometry. Three-dimensional (3D) echocardiography may overcome some of these limitations but is not yet widely available [2]. In addition, the LVEF is not sensitive enough to detect minor changes, which is a limitation for the early detection of cardiotoxicity. Thus, a decrease in the LVEF represents a relatively late stage of LV systolic function impairment.

Strain by speckle tracking echocardiography (STE) is a technique that utilizes 2D gray-scale images to evaluate both the global and regional functions of the LV. Global longitudinal strain (GLS) from 2D STE is an emerging method for detecting and quantifying subtle disturbances in global long-axis LV systolic function. It is operator-independent, more reproducible than the LVEF, and easily measured and integrated with standard echocardiography. Myocardial strain reflects changes in tissue deformation during each cardiac cycle and is referenced to the original length. Since cardiomyocytes shorten during systole, this is described as a negative number, but when the strain is averaged in all myocardial segments (GLS), it is always negative, meaning it can be reasonably expressed without a negative sign (i.e., global longitudinal shortening) [3].

The global longitudinal strain may be used to measure the LV systolic function [4,5,6]. There are many studies on normal GLS values in healthy volunteers. Data from the latest meta-analysis suggest that GLS LV values >18% should be considered normal and values between 16% and 18% as borderline, in part because of the load-dependence of the strain [7]. Changes in loading conditions may affect not only the cardiac volumes and LVEF but also the quantification GLS. In addition, it needs underlying that strain measurements may be subject to inter-vendor variability. Thus, serial GLS measurements should be performed for each patient using the same machine and software.

### 1.1. Definition of Cardiotoxicity

Cardiotoxicity has most often been defined as an absolute decrease in 2D LVEF by >10% from the baseline and up to <53%. According to another postulated definition, cardiotoxicity should be diagnosed in the case of a decrease in the LVEF by >10 absolute percentage points to a value of <50%. The probable cardiotoxicity is reflected by a decrease in the LVEF by >10 absolute percentage points to a value of ≥50% with an accompanying decline in GLS by >15% from the baseline (if GLS measurement is available). Possible cardiotoxicity by echocardiography is recognized when the LVEF decreases by <10 absolute percentage points to a value of <50% or when the LVEF ≥ 50%, in the case of a relative percentage reduction in GLS by >15% [8]. The GLS is well established in these definitions, as a relative decrease in GLS of >15% from baseline is considered an indicator of subclinical LV dysfunction [9].

The GLS is an important element in diagnosing the cardiotoxicity of cancer treatment, and the latest ESC guidelines on cardio-oncology define three degrees of cardiotoxicity [10]. Severe cardiotoxicity is indicated by a new LVEF decline to <40%. A new LVEF decrease by ≥10 percentage points to an LVEF of 40–49%, or a new LVEF decline by <10 percentage points to an LVEF of 40–49%, and either a new relative decline in GLS by >15% from the baseline or a new increase in cardiac biomarkers are indicators of moderate cardiotoxicity. Mild cardiotoxicity can be diagnosed when the LVEF is ≥50 with a new relative decline in GLS by >15% from the baseline or a new increase in cardiac biomarkers.

### 1.2. GLS in the Diagnosis of Cardiotoxicity

GLS has been studied to detect early changes in LV contractile function in patients undergoing cardiotoxic chemotherapy. Due to the simplicity and repeatability of its measurements, many authors have attempted to determine a cut-off for GLS that could indicate the occurrence of cardiotoxicity. According to Gripp et al., a 14% reduction in strain (absolute value of −16.6) facilitated the early identification of patients who may develop anthracycline- or trastuzumab-induced cardiotoxicity [11]. A meta-analysis by Oikonomou et al. confirmed that GLS can be a valuable tool for the early detection of cardiotoxicity associated with oncological treatment, with the caveat that the studies published so far have shown significant statistical heterogeneity and that larger prospective multicenter studies on this topic are needed [12]. In this meta-analysis of 21 prognostic studies, which included patients treated with anthracyclines, with or without trastuzumab, the authors indicated absolute changes in GLS between 2% and 3% and relative changes of between 10% and 15% as the cut-off points that permit the diagnosis of subclinical cardiotoxicity, with a sensitivity range of 80–90% and a specificity rate of 80%. Similar data were provided by a recent meta-analysis published by Cocco et al. [13]. Patients diagnosed with cardiotoxicity had a 14.13% greater strain reduction from the baseline than those without cardiotoxicity. This meta-analysis showed that GLS is a tool with adequate predictive capacity for the detection of cardiotoxicity and LV subclinical dysfunction [13].

In addition, baseline GLS or decreased baseline GLS was shown to be a predictor of cardiotoxicity in a cohort of cancer patients with normal baseline LVEF who underwent treatment with anthracyclines [14]. Other authors have shown that patients with preserved GLS (≤−17%) had a significantly lower risk of developing cardiotoxicity, and for every 1-unit improvement in GLS, the risk of cardiotoxicity was reduced by 16% [15]. Baseline determinations of LVEF and GLS are, therefore, recommended in all patients prior to the initiation of potentially cardiotoxic anticancer treatments in order to stratify the risk and identify significant changes during therapy [10].

Unfortunately, despite the evidence that GLS is a sensitive marker of subclinical cardiotoxicity, it has not been confirmed that the strategy based on GLS assessment was superior to LVEF monitoring during anticancer therapy. SUCCOUR was the first multicenter prospective study of patients receiving cardiotoxic chemotherapy and undergoing echocardiographic surveillance of LV function, who were randomly allocated to a GLS-guided arm and LVEF-guided arm with a 1:1 ratio [16]. Patients treated with anthracyclines were included in the study, and other risk factors for cardiotoxicity were also taken into account in the analysis (additional potentially cardiotoxic chemotherapy, including trastuzumab, tyrosine kinase inhibitors, or high-dose of anthracycline, and risk factors for cardiovascular diseases, including heart failure). They were to be followed over 3 years for the primary endpoint (change in 3D LVEF) and other secondary endpoints. The cardioprotective treatment was started when the study criteria for cancer-therapy-related cardiac dysfunction were met. In the conventional imaging (LVEF-guided group), this was defined by a symptomatic drop of >5% of the LVEF or a >10% asymptomatic drop to LVEF < 55%. In the GLS-guided group, this was defined by a relative reduction in GLS by ≥12% as compared with the baseline. The results of the study, which enrolled 331 subjects, were disappointing; the primary endpoint was not met because the difference in LVEF values between the groups at 1-year follow-up was not statistically significant (GLS-guided group 57% ± 6% vs. LVEF-guided group 55% ± 7%; *p* = 0.05) [17]. Whether or not a relative reduction in GLS of 12% is sufficient to diagnose subclinical cardiotoxicity and to identify patients who would benefit from cardioprotective treatment to prevent cancer-therapy-related LV dysfunction, it should be emphasized that strain impairment does not always indicate LV dysfunction given its preload dependency [18]. In addition, it was observed that in patients at low risk for cardiotoxicity, GLS alterations may be reversible and not associated with clinically significant cardiotoxicity or a late LVEF decrease [19].

### 1.3. Assessment of GLS in Combination with Cardiac Biomarkers

The assessment of biomarkers indicating myocardial damage or overload may increase the importance of an abnormal strain finding. Avila et al. showed that the association of low GLS values < 17% and brain natriuretic peptide (BNP) serum concentrations > 17 pg/mL two months after chemotherapy increased the accuracy for detecting early-onset cardiotoxicity (100% sensitivity, 88% specificity, AUC = 0.94) [20]. Sulaiman et al. examined female asymptomatic breast cancer patients and also found that a combined relative reduction in GLS and the relative elevation of the N-terminal pro B-type natriuretic peptide (NT-proBNP) concentration allowed for defining subtle subclinical anthracycline-induced cardiotoxicity as early as after 6 weeks from its first dose [21]. Not only natriuretic peptides but also troponins are considered important cardiac biomarkers of cardiotoxicity, which is why some authors suggest a combined assessment of GLS with serum troponin levels. It was reported that a >15.9% decrease in GLS and a >0.004 ng/mL elevation in the highly sensitive cardiac troponin T (cTnT) concentration after the third cycle of chemotherapy with epirubicin predicted later cardiotoxicity [22]. An interesting scoring system, the CardTox-Score, has been recently proposed for predicting non-anthracycline chemotherapy-induced cardiotoxicity. The variables of this risk model, apart from GLS values < −20%, consist of clinical data (age ≥ 60 years, BMI > 25 kg/m^2^, presence of cardiovascular risk conditions such as arterial hypertension, diabetes, smoking, dyslipidemia, coronary artery disease), laboratory markers (highly sensitive troponin I > 0.04 pg/mL, NT-proBNP > 400 pg/mL), and echocardiographic variables (LVEF ≤ 50%, LV diastolic dysfunction ≥ grade 1). A CardTox-Score > 6 points was identified as a strong independent predictor for the development of subsequent cardiotoxicity, with a sensitivity rate of 100% and specificity rate of 84.2%. This scoring system seems to be a useful tool for predicting the risk of chemotherapy-induced cardiac toxicity in oncological patients undergoing non-anthracycline anticancer regimes, independently of the type of cancer [23].

### 1.4. GLS vs. Segmental Strain

The single versus standard multi-view assessment of GLS was examined when looking for optimal methods for cardiotoxicity assessment using speckle tracking echocardiography. The single-view longitudinal strain measurement can lead to disagreement in the diagnosis of cardiotoxicity in up to 22% of patients. Therefore, a GLS assessment based on 3 apical views should remain the preferred method for the detection of cardiotoxicity [24]. Some authors, when assessing the longitudinal deformation of the LV, went a step further, trying to define which segments most often present contractile dysfunction. Portugal et al., assessing cardiotoxicity among breast cancer patients, showed that predominantly the septal and anterior walls were involved [25]. In another study on breast cancer patients during treatment with fluorouracil, doxorubicin, and cyclophosphamide, the greatest regional longitudinal strain decline was detected in the antero-apical segment [26]. In turn, Mahjoob et al. found that decreased lateral and infero-septal segmental longitudinal strain were specific markers of anthracycline-related cardiac toxicity [27]. These data indicate that the cardiotoxicity of chemotherapy is a complex problem, and at present it is not possible to identify one or more LV segments in which the segmental reduction in longitudinal strain could be a reliable marker of cardiotoxicity of oncological therapy. Arciniegas-Calle et al. showed that anthracycline–trastuzumab treatment leads to early deterioration not only in the LV GLS, but also in the global circumferential strain, LV systolic strain rate (SR), right ventricular (RV) GLS, and RV SR [28]. However, the data are currently insufficient to recommend their use routinely.

### 1.5. The Use of GLS Assessment in Patients with Various Cancers

Most of the studies on GLS in the diagnosis of subclinical cardiotoxicity concern patients with breast cancer treated with anthracyclines or HER2-targeted therapies (mainly trastuzumab). However, it was shown that GLS could detect subtle but clinically significant cardiac dysfunction in lymphoma patients in the early stage of anticancer therapy [29]. In addition, baseline GLS was the predictor of LV dysfunction and hospitalization for heart failure in patients with malignant lymphoma after anthracycline therapy. An ROC analysis identified the GLS cut-off for predicting LV dysfunction after anthracycline chemotherapy as ≤19% [30]. Gonzalez-Manzanares et al. have evaluated childhood leukemia survivors in terms of cardiotoxicity, comparing conventional echocardiography and automated software that simplifies the GLS measurements. Their results confirmed that automated GLS measurements are superior to conventional echocardiography in the early detection of cardiotoxicity [31]. There are, however, reports on GLS assessments in patients with other cancers undergoing other oncological therapies. Interesting data were obtained by Oka et al., who examined serial changes in cardiac strain and contractility in patients with hematologic malignancies after hematopoietic stem cell transplantation (HSCT). They found that GLS before HSCT might be associated with a decrease in LVEF after HSCT [32]. The results from a study on patients with colorectal cancer confirmed the usefulness of GLS in the diagnosis of bevacizumab-induced cardiotoxicity [33]. LV GLS assessments have also proven useful in patients with multiple myeloma (MM). Therapy with proteasome inhibitors, e.g., carfilzomib, may be cardiotoxic, as demonstrated by GLS changes and diastolic dysfunction occurrence [34,35]. In addition, the thickness of the LV muscle is important in these patients. It was shown that the LV myocardial global work index, myocardial global work efficiency, and GLS were lower in patients with MM and thick walls compared to patients with normal walls [36]. There have also been reports on GLS in assessing the prognosis in non-small cell lung cancer (NSCLC) patients. The percentage change in GLS from baseline to 6 months after radiotherapy was an independent predictor of all-cause mortality. Based on the ROC analysis, a relative reduction in GLS of ≥13.65% was the cut-off value for predicting mortality in NSCLC patients. These findings should encourage physicians to perform echocardiography early after radiotherapy [37].

The GLS was also examined in patients with advanced light-chain (AL) amyloidosis. Baseline GLS appeared to be an independent predictor of overall survival in these patients. Those with GLS values < −14.2% had corresponding median overall survival (OS) and 5-year OS rates of 33.2 months and 39%, respectively, versus 7.7 months and 6% for those with GLS values ≥ −14.2% [38].

### 1.6. GLS Depending on the Method of Anticancer Treatment

GLS has been shown to be a valuable diagnostic tool in detecting the cardiotoxicity of various methods used for cancer treatment, not just chemotherapy. A decline in GLS is observed in patients with immune checkpoint inhibitor (ICI)-related myocarditis, and lower GLS was strongly associated with major adverse cardiac events (MACE), irrespective of the LVEF [39]. LV dysfunction during ICI therapy does not necessarily result from myocarditis. GLS monitoring in melanoma patients can detect ICI-induced subclinical LV dysfunction (in the absence of myocarditis) [40]. In addition, an early relative worsening of ≥10% in the basal and mid-longitudinal strain and ≥15% in GLS was associated with hsTnI elevation among patients treated with ICIs [41]. GLS may also reveal the cardiotoxicity of radiotherapy in its subclinical stage. In breast cancer patients treated with left-sided radiotherapy, the radiation dose correlated with a subclinical reduction in the segmental longitudinal strain, and a limit of 5 Gy in LV segments should be considered during radiotherapy planning [42]. Additionally, a BACCARAT study indicated that subclinical LV dysfunction, defined as a GLS decrease of >10%, was associated with cardiac doses. The patients at significant risk of developing subclinical LV dysfunction were those with a relative LV volume exposed to at least 20 Gy of >15% (LV V20 > 15%) [43]. In a recent meta-analysis, Xu et al. summarized reports on the use of GLS in the assessment of later cardiac complications after radiotherapy. GLS has been shown to be a good parameter to detect early radiation-induced heart disease in women with left breast cancer. However, in the case of right breast cancer, segmental changes may be more important [44].

### 1.7. GLS Assessment in the Context of Right Ventricular Cardiotoxicity

Keramida et al. showed that the deformation mechanics of both LV and RV follow similar temporal patterns and grades of dysfunction during trastuzumab therapy, confirming the global and uniform effect of anty-HER2 drugs on cardiac function. They proposed a cut-off value of −14.8% for the RV GLS percent change, which discriminated against patients with trastuzumab-related cardiotoxicity. It was similar to the LV GLS change (−15%) postulated as a cut-off for the diagnosis of subclinical cardiotoxicity. An RV GLS percent change of −14.8% predicted cardiotoxicity with a sensitivity rate of 66.7% and a specificity rate of 70.8% (AUC 0.68, 95% confidence interval 0.54–0.81), correctly identifying 90% of women with cardiotoxicity [45]. Some authors suggest that cardiotoxicity after anthracyclines in the form of worsened RV GLS is even more common than lessened LV GLS [46]. Other authors, evaluating post-anthracycline cardiotoxicity in children treated for osteosarcoma, also confirmed the importance of RV GLS assessment in the diagnosis of cardiotoxicity, suggesting that the RV function decreases early, even before LV dysfunction [47].

### 1.8. GLS Assessment during Cancer Treatment 

What is extremely useful in the latest ESC guidelines on cardio-oncology is that depending on the baseline risk of cardiotoxicity, they specify how often follow-up echocardiography should be performed during the use of various potentially cardiotoxic cancer treatments [10]. This has been discussed in detail in relation to anthracyclines and HER2-targeted therapies; for VEGF, BCR-ABL, and Bruton tyrosine kinase inhibitors; for proteasome inhibitors, MEK and RAF inhibitors, and immune checkpoint inhibitors; and during follow-up after radiotherapy involving the mediastinum and the left chest. The recent recommendations from scientific societies on this subject are presented in Table 1, along with a comparison of the suggested time intervals. 

The importance of GLS in monitoring patients receiving potentially cardiotoxic chemotherapy and in diagnosing cardiotoxicity at an early, subclinical stage is demonstrated in the study by Di Lisi et al. The authors assessed the impact of the COVID-19 pandemic on the occurrence of cardiac dysfunction associated with anticancer treatment. It was confirmed that during the COVID-19 pandemic, cardiac surveillance was severely limited, follow-up visits were kept to a minimum, and GLS estimations were often omitted from echocardiographic examinations. According to the authors, this was one of the main reasons for the higher incidence of cancer-therapy-related cardiac dysfunction after the third wave of the COVID-19 pandemic as compared to the same period in 2019 before the COVID-19 pandemic [52]. 

### 1.9. Perspectives of GLS Assessment

Despite the well-established role of GLS in cardio-oncology, new solutions for STE analysis are constantly sought, which will enable more effective diagnostics of cardiotoxicity resulting from oncological treatment. High hopes are pinned on 3D GLS evaluations. In a study by Piveta et al. in breast cancer patients, the 3D GLS assessment indicated early strain changes after very low doses of anthracyclines. These changes were also associated with a subsequent decrease in LVEF [53]. Intensive work is also underway to apply artificial intelligence to a fully automatic GLS assessment, which will minimize the variability of the results obtained by different investigators and between software programs from different manufacturers of echocardiographic equipment [54]. In addition, GLS calculations based on MRI and with the use of artificial intelligence have also been shown to be early predictors of cardiotoxicity in breast cancer patients [55]. It should also be emphasized that such a parameter as myocardial work, which is a derivative of GLS, may be of added value in the diagnosis of cardiotoxicity. Providing a measure of myocardial work (independent of LV afterload) may be an important advance in the monitoring of patients undergoing potentially cardiotoxic oncological treatments [56].

### 1.10. Disadvantages of GLS

One of the conditions for a reliable assessment of GLS is obtaining good quality echocardiographic images. In the case of suboptimal conditions, the inter-observer reproducibility of GLS measurements deteriorates significantly. In patients with breast cancer, the quality of the images obtained is especially affected by left-sided mastectomy or reconstructive surgery and a high body mass index [57]. 

Since the GLS is load-dependent, variation in the sequential imaging results may be due to differences in blood pressure; lower GLS values may be attributed to higher blood pressure values at the time of measurement [58].

Learning the GLS assessment is relatively simple; it is estimated that the performance of a minimum of 50 tests results in obtaining the competence of an expert. On the other hand, the short-axis strain analysis of global circumferential strain is much more difficult. 

Hopes related to the use of GLS in cardio-oncology were shattered by the results of the already mentioned SUCCOUR study. This well-designed study provides the best evidence yet that GLS does not play a significant role in the routine surveillance of patients treated with potentially cardiotoxic anticancer therapy [16,59]. This was indicated by the results of both the 12-month and 3-year follow-up in this study. After 3 years, an improvement in LV dysfunction was demonstrated compared to the results obtained after 12 months. However, there were still no differences in the LVEF values between the two compared groups of patients, including between patients in whom cardioprotection was initiated based on GLS monitoring and those in whom LVEF was assessed [60]. In the SUCCOR study, however, a milder criterion was used—GLS deterioration by 12% as an indication for cardioprotection. However, the recommendations of scientific societies indicate a 15% decrease in GLS, which may be due to subclinical cardiotoxicity.

## 2. Conclusions

The GLS is valuable in diagnosing patients at risk for LV systolic dysfunction and in monitoring patients undergoing potentially cardiotoxic anticancer therapy. Due to its simplicity and repeatability, the GLS assessment has been rapidly incorporated into cardio-oncology (Figure 1). A relative decrease in GLS of >15% compared to baseline during cancer treatment is the recommended cut-off point for suspecting subclinical cardiac dysfunction and predicting a significant LVEF decrease in the future. An early diagnosis of asymptomatic cardiotoxicity enables the quick implementation of a cardioprotective treatment and reduces the risk of interruptions or changes in the oncological treatment, which may affect the survival of patients. Additional research is needed to precisely assess both the LV GLS and RV GLS in the constantly evolving field of cardio-oncology.

## Figures and Tables

**Figure 1 cancers-15-00986-f001:**
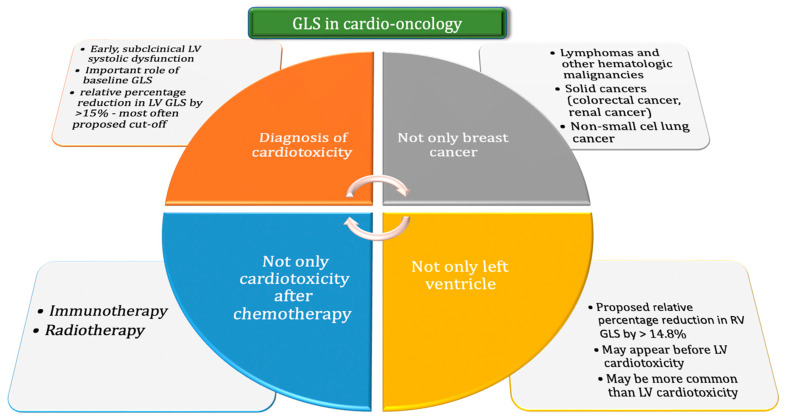
The use of GLS in cardio-oncology, taking into account the role of GLS in the diagnosis of cardiotoxicity, the possibility of assessing this parameters in many types of cancer and for many types of oncological treatment techniques, and the possibility of assessment also for the right ventricle. Abbreviations: GLS = global longitudinal strain; LV = left ventricular; RV = right ventricular.

**Table 1 cancers-15-00986-t001:** Recommendations of scientific societies in recent years regarding the frequency of performing an echocardiographic examination with a GLS assessment.

Guidelines, Year	Anthracyclines	HER2-Targeted Therapies
2022 ESC Guidelines on cardio-oncology developed in collaboration with the European Hematology Association (EHA), the European Society for Therapeutic Radiology and Oncology (ESTRO), and the International Cardio-Oncology Society (IC-OS): Developed by the task force on cardio-oncology of the European Society of Cardiology (ESC) [10]	Low baseline cardiotoxicity risk: at baseline and 12 months post-treatment (may be also considered after the fourth cycle) Moderate risk: at baseline and 12 months posttreatment (should be also considered after the fourth cycle). High and very high risk: at baseline, after 2nd, 4th, and 6th cycles and also 3 and 12 months after treatment	Low and moderate risk: at baseline; after 3, 6, 9, and 12 months; and then 12 months post-treatment High and very high risk: at baseline; after 3, 6, 9, and 12 months; and then 3 and 12 months post-treatment
2021 British Society for Echocardiography and British Cardio-Oncology Society guideline for transthoracic echocardiographic assessment of adult cancer patients receiving anthracyclines and/or trastuzumab [8].	Every 3 months during chemotherapy, and 3–12 months after its termination	Every 3 months during the therapy and 3–12 months after the end of therapy
2020 Management of cardiac disease in cancer patients throughout oncological treatment: ESMO consensus recommendations [48].	After a cumulative dose of 250 mg/m^2^ doxorubicin or equivalent. Next after each additional 100 mg/m^2^	Every 3 months
2020 Role of cardiovascular imaging in cancer patients receiving cardiotoxic therapies: a position statement on behalf of the Heart Failure Association (HFA), the European Association of Cardiovascular Imaging (EACVI), and the Cardio-Oncology Council of the European Society of Cardiology (ESC) [49].	Depending on the risk calculated according to the planned therapy and patient-related risk factors, including age, comorbidities, and cardiovascular (CV) risk factors	Risk is calculated according to the planned therapy and patient-related factors, including age, comorbidities, and CV risk factors (range from 6 to 12 weeks)
2016 Canadian Cardiovascular Society Guidelines for Evaluation and Management of Cardiovascular Complications of Cancer Therapy [50].	No recommendations	Every 3 months
2016 ESC Position Paper on cancer treatments and cardiovascular toxicity developed under the auspices of the ESC Committee for Practice Guidelines: The Task Force for cancer treatments and cardiovascular toxicity of the European Society of Cardiology (ESC) [51].	After a cumulative dose of 200 mg/m^2^ doxorubicin or equivalent	Every 4 cycles

## Data Availability

No new data were created or analyzed in this study. Data sharing is not applicable to this article.

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
