# Peer review of "Global Longitudinal Strain in Cardio-Oncology: A Review"

_cancers, 2023, doi:10.3390/cancers15030986_

Round 1
Reviewer 1 Report
The authors have investigated and important cardiovascular theme regarding cardio-oncology. Echocardiography and its techniques are of great use in early detection of cardiac damage. To this regard Global Longitudinal Strain represents a feasible technique to be enhanced in the cardio-oncology reality. This review highlights the state of the art of this technique and the relative cardio-oncology recommendations from the most authoritative guidelines.
The study is interesting, well thought of and overall well written. Introduction is sufficient and on point. I appreciate the point by point focus on the various GLS aspects. The table and the figure are of universal interpretation.
My minor revision consists in the need in the conclusion for future prospective. This review should recommend where to look towards in this field of application.
Author Response
The authors have investigated and important cardiovascular theme regarding cardio-oncology. Echocardiography and its techniques are of great use in early detection of cardiac damage. To this regard Global Longitudinal Strain represents a feasible technique to be enhanced in the cardio-oncology reality. This review highlights the state of the art of this technique and the relative cardio-oncology recommendations from the most authoritative guidelines.
The study is interesting, well thought of and overall well written. Introduction is sufficient and on point. I appreciate the point by point focus on the various GLS aspects. The table and the figure are of universal interpretation.
My minor revision consists in the need in the conclusion for future prospective. This review should recommend where to look towards in this field of application.
Answer:
Thank you very much for this review. As suggested, we have added an additional paragraph on the perspectives of GLS in cardio-oncology, where we inform about the use of 3DGLS, GLS from MRI, or the use of artificial intelligence in analysis. We have added relevant references.
Reviewer 2 Report
The authors have done a review of the published literature on global longitudinal strain. The review is well written. A few suggestions are made here.
1. The headings can be better, especially for 1.5, 1.6, and 1.7- The current section headings do not help understand the content.
2. Please consider talking about myeloma therapies and checkpoint inhibitors separately.
Author Response
The authors have done a review of the published literature on global longitudinal strain. The review is well written. A few suggestions are made here.
- The headings can be better, especially for 1.5, 1.6, and 1.7- The current section headings do not help understand the content.
Answer: Thank you for this comment. We changed the indicated headings to new ones, better reflecting the content of individual paragraphs.
- Please consider talking about myeloma therapies and checkpoint inhibitors separately.
Answer: Thank you for this comment. We expanded the paragraphs describing the use of GLS among patients with multiple myeloma and among patients treated with ICIs.
Reviewer 3 Report
Interesting article. However there are few improvements can be done.
I advise the article to be sent to professional English editing service as I feel there are improvement on the language aspect. Please check grammar eg. Line 40. Line 103 word “strain” should be GLS, right? Line 107- what is preserved GLS? please describe.
Line 212- detecting cardio toxicity and other methods? - check sentence.
42 references for this review. No reference in the first paragraph. A good review should have sufficient number of references to show that the review is well supported by the literature.
I feel the author contribution be made clearer eg. who wrote the first draft, who edited etc.
Author Response
Interesting article. However there are few improvements can be done. I advise the article to be sent to professional English editing service as I feel there are improvement on the language aspect. Please check grammar eg. Line 40.
Answer: Thank you for these comments. We proofread the manuscript once again, corrected linguistic errors.
Line 103 word “strain” should be GLS, right?
Answer: That's right. Corrected.
Line 107- what is preserved GLS? please describe.
Answer: In this case, the phrase "preserved GLS" referred to patients from the cited study. In this study, the authors divided patients into two groups of patients: reduced GLS (>−17%) vs. preserved GLS (≤−17%). We have added information about this cut-off point in the text. lines:
Line 212- detecting cardio toxicity and other methods? - check sentence.
Answer: Fixed a linguistic error.
42 references for this review. No reference in the first paragraph. A good review should have sufficient number of references to show that the review is well supported by the literature.
Answer: We agree that a good quality review should contain the right amount of references. Therefore, we added an additional 11 references, resulting in a total of 53 references.
I feel the author contribution be made clearer eg. who wrote the first draft, who edited etc.
Answer: Thank you for this comment. An editorial error has crept in. Of course, we added contributions.
Round 2
Reviewer 1 Report
The authors have made the required adjustments. The article is now fit for publication.